# Trait Energy and Fatigue Modify Acute Ingestion of an Adaptogenic-Rich Beverage on Neurocognitive Performance

Ali Boolani [1,2,*], Daniel Fuller [3], Sumona Mondal [3] and Eric Gumpricht [4]

1 Department of Physical Therapy, Clarkson University, Potsdam, NY 13699, USA
2 Department of Biology, Clarkson University, Potsdam, NY 13699, USA
3 Department of Mathematics, Clarkson University, Potsdam, NY 13699, USA; fullerdt@clarkson.edu (D.F.); smondal@clarkson.edu (S.M.)
4 Isagenix International, LLC, Gilbert, AZ 85297, USA; eric.gumpricht@isagenixcorp.com
* Correspondence: aboolani@clarkson.edu

**Abstract:** Background: Psychological research considers traits as a long-standing pre-disposition to an individual's mood, whereas short-term feelings are categorized as states. We previously reported similar overall acute mental performance benefits between an adaptogen-rich, caffeine-containing energy shot (e+Energy Shot–e+Shot; Isagenix International, LLC) and a caffeine-matched placebo Since the publication of that study, multiple studies have reported that trait mental and physical energy (TME/TPE), and trait mental and physical fatigue (TMF/TPF) status modify the effect of various interventions on neurocognitive performance. Therefore, we reevaluated our previously published work and accounted for the four traits. Methods: Participants (*n* = 30) completed a series of questionnaires to determine baseline trait energy and fatigue measures. Then, participants performed a 27 min battery of neurocognitive tasks before and three times after consuming the study beverages with 10 min breaks between each post-consumption battery of tests. Data from the previous study were re-analyzed using linear mixed-effects models. Results: We now report that the adaptogen product significantly improved mood and cognitive test responses in individuals stratified by initial TME, TPE, TMF, and TPF status. Moreover, this reevaluation also indicated that the caffeine placebo significantly increased heart rate and blood pressure in those subjects initially characterized by low physical and mental energy. Conclusions: In summary, a post-hoc re-analysis of our initial study suggests that consumption of the adaptogen-rich, caffeine-containing product preferentially benefited individuals with initial low TME/TPE and high TMF status when compared to caffeine alone. These findings also support our previous study suggesting that adaptogens may promote mental and physical performance benefits while modulating potentially negatively associated responses to caffeine.

**Keywords:** trait energy; trait fatigue; adaptogens; caffeine; cognition; energy; fatigue; mood

## 1. Introduction

In psychological research, trait moods are considered long-standing pre-dispositions to a mood [1]. In contrast, short-term, transient feelings are defined as states [1]. Traits and states also interact with each other with trait moods influencing state mood frequency and intensity [2–4]. For example, someone who feels typically depressed (high trait depression) reports feeling depressed more frequently and more intensely [5].

Recently, a study from our lab suggest trait mental and physical energy (TME/TPE), and trait mental or physical fatigue (TMF/TPF) modify the influence of sleep on state energy and fatigue [4]. Other studies from our lab also report that TME, TPE, TMF, and TPF modify the effects of state energy and fatigue on postural control [6], and gait [6,7] in young adults, as well as functional outcomes in older adults in an aquatic settings [8]. Additionally, dietary interventions from our lab also suggest that TME, TPE, and TMF modify caffeine's effects on neurocognitive performance [9]. Specifically, we reported that trait status influenced

the intensity of mood changes. For example, individuals exhibiting high TME or TPE or low TMF revealed the most significant benefits of consuming caffeine on subjective and objective indices of mental energy. One potential explanation for these interactions between caffeine and traits may be attributed to the gut microbiome and functional pathways of trait energy and fatigue [10] potentially influencing caffeine metabolism [11].

Other research suggests interactions between trait status on individual genotypes. For example, researchers have reported that the genetic polymorphism of COMT rs9332377 was associated with fatigue intensity [12] and a genetic polymorphism that modifies catecholamine metabolism in response to stress [13]. Strong evidence also reveals inter-individual differences in caffeine's influence upon mood, cognitive task performance, and psychomotor responses [12,14]. Accordingly, identifying additional dietary, genetic, or other factors (i.e., trait moods) that may explain inter-individual differences in caffeine's psychophysiological responses may be a particularly fruitful endeavor considering the potential beneficial effects of caffeine.

Indeed, this trait interaction on the neurocognitive effects of caffeine is similar to what we have recently reported [9]. The study's findings led us to re-examine our previous data examining the effects of two different caffeinated beverages on neurocognitive parameters after the performance of psychologically fatiguing tasks [15]. In our previous study [15], we evaluated the effects of a synthetically caffeinated beverage vs. a beverage providing similar caffeine content along with the addition of a proprietary adaptogenic herbal blend (e+Energy Shot^TM—Isagenix International, LLC; Gilbert, AZ, USA). Adaptogens comprise a category of plants long utilized as traditional medicine by many cultures over millennia, are characterized by a strict inclusion criterion, and first systematically studied by scientists in the Soviet Union [16,17]. These plants thrive in harsh environmental climates (hence the term, "adaptogen") and have been long-valued for their purported benefits towards mitigating stress-induced systematic damage or improving general health and well-being via numerous, non-specific cell signaling and modification of the hypothalamus–pituitary–adrenal (HPA) axis. We initially reported relatively similar outcomes on objective and subjective mental energy and fatigue measures between the beverages. For example, when participants consumed the caffeinated beverage, there was a spike 30 min post-consumption for most objective and subjective measures of mental energy and mental fatigue before tapering off. In contrast, the e+Energy Shot (e+shot) beverage reported a steady improvement over a more extended period. However, new findings suggesting a significant and complex relationship between one's trait status and response to psychophysiologically active dietary components (such as caffeine and adaptogenic herbs) have led us to perform a post-hoc analysis of our previous data [15].

## 2. Methods

### 2.1. Study Design

In our previous study [15], we compared the cognitive and physiological effects among three interventions: (1) a placebo; (2) e+shot; (3) a caffeinated-matched active comparator (caffeine placebo). A complete and detailed description of the methodology was previously published [15]. Here, we solely compared the active comparator (caffeine) vs. e+shot. To confirm participant compliance and blinding, all beverages were provided via visually identical and unmarked bottles. All researchers involved with the study or analysis were also blinded to interventions and their allocations. The caffeinated contained approximately 98 mg synthetic caffeine and consisted of the same base components as e+shot. In contrast, the caffeine in e+shot (approximately 85 mg) was provided by a combination of green tea (*Camelia sinensis*) and yerba mate (*Ilex paraguariensis*). Table 1 lists the adaptogenic herbs in e+shot and Table 2 provides an overall comparison between the two beverage shots. Eurofins Scientific Inc. (Des Moines, IA, USA) verified the quantification of caffeine in both study products.

**Table 1.** Adaptogenic Herbs Contained in e+Shot (mg).

| | |
|---|---|
| Siberian Ginseng (*Eleutherococcus senticosus*) | 79 |
| Hawthorn (*Crataegus oxycantha*) | 59 |
| Mountain ash (*Sorbus aucuparia*) | 59 |
| Cramp bark (*Viburnum opolus*) | 59 |
| Leuzea (*Rhaponticum carthamoides*) | 40 |
| Rhodiola (*Rhodiola rosea*) | 20 |
| Japanese aralia (*Aralia mandchurica*) | 20 |
| Licorice (*Glycyrrhiza yuralensis*) | 20 |
| Schizandra (*Schisandra chinensis*) | 20 |
| Chaga mushroom (*Inonotus obliquus*) | 20 |

**Table 2.** Comparison of Study Product Bioactives.

| Treatment | Caffeine (mg) | Adaptogenic Herbal Blend (mg) |
|---|---|---|
| Placebo | 0 | 0 |
| Active Comparator (Caffeine) | 98 (synthetic) | 0 |
| e+Shot | 85.4 (green tea-*Camellia sinensis* and yerba mate-*Ilex paraguariensis* leaf extract) | 2127 |

*2.2. Measures*

Prior to study acceptance, individuals completed several assessments. In line with our study aims, these individuals provided information pertaining to objective and subjective measures of mental parameters and mood. The full description of pre-testing and testing day measures' narratives could be found in *Blinded* [18] and *Blinded* [15], respectively.

All sessions collected subject data in a seated position in a temperature-controlled environment (72 ± 0.8 °F/22.2 ± 0.8 °C); the setting also included carefully controlled sound and lighting. A 17" laptop screen exhibited targeted visual stimuli and participants were asked to use the keyboard to comply with the instructions provided. All other details are provided in our prior publication [15].

*2.3. Pre-Testing Measure*

For this analysis, we considered the trait aspect of mental and physical state and trait energy and fatigue scale (MPSTEF). Additionally, with the MPSTEF we collected pre-testing data on trait predisposition, as well as data on mental and physical energy and fatigue. In this context, the MPSTEF consists of 12 categories corresponding with three items each pertaining to four possible trait outcomes. As an example, statements may include: "I feel I have energy" and "I have feelings of being worn out." Responses were presented on a 5-point Likert scale ranging from "never" (0) to "always" (4). In comparison, analysis from similar-type studies revealed a Cronbach's $\alpha$ coefficient in the range of 0.82 to 0.93 [4,18,19]. The Cronbach $\alpha$ coefficient for the current study exhibited a range between 0.73 and 0.88 (TME = 0.73, TMF = 0.88, TPE = 0.75, and TPF = 0.84).

*2.4. Screening and Participants*

Upon Institutional Review Board (IRB) approval (approval #16-34.1), recruitment for the study was undertaken from within and outside the university via a combination of efforts, including: university announcements, bulletins, electronic listservs, advertisements at local businesses, and personal communication, such as word of mouth. Interested individuals completed an online Survey Monkey (SurveyMonkey.com) screening questionnaire.

The exclusion criteria for this study were considerable, and included: children under 18 years or adults over 45 years; individuals classified as obese with a body mass index (BMI) > 30; elevated feelings of energy per a 30 item Profile of Mood Survey-Short Form (POMS-SF) (scores > 12); individuals with elevated caffeine consumption (>21 servings/>341 mL caffeine beverages/week); individuals with any chronic health ailment requiring a consistent medication usage (excluding contraception); pregnancy or potential pregnancy during the intervention testing periods; a self-reported caffeine allergy; current smoker; regular usage of any dietary nutritional supplements (i.e., herbs, vitamins, or creatine), excluding caffeine-free protein supplementation.

Individuals fulfilling these criteria completed an informed consent, and were notified of their participation eligibility for an investigation into the mental and physiological effects of herbal and caffeine-containing beverage shots. Characteristics of the thirty (women = 17, men = 13) participants, including all the post-hoc analyses, are reported in Table 3.

**Table 3.** Participant Characteristics.

| | |
|---|---|
| Sex (Males/Females) | 13/17 |
| Age (years) | $21.8 \pm 4.4$ |
| Height (cm) | $169.6 \pm 12.4$ |
| Weight (kg) | $67.6 \pm 11.0$ |
| Body Mass Index (kg/m$^2$) | $23.5 \pm 2.5$ |
| Race | |
| White | 21 |
| Asian | 4 |
| Black | 4 |
| More than one race | 1 |
| Amount of sleep on a typical night in the past month (h) | $7.6 \pm 0.8$ |
| Consumption of high-flavanol foods or beverages during the past month | |
| Caffeine drinks (servings) | $4.2 \pm 3.8$ |
| Cocoa (servings) | $0.7 \pm 1.3$ |
| Fruits (servings) | $12.3 \pm 12.4$ |
| Vegetables (servings) | $25.1 \pm 14.5$ |

The average reported duration of sleep the month prior to the interventions was $7.6 \pm 0.8$ h. Additionally, sleep durations the night prior to testing sessions were similar (t = $-0.433$, $p = 0.666$) between the caffeine ($6.4 \pm 1.1$ h), and the e+shot ($6.3 \pm 1.2$ h) groups. Overall, participants were considered low caffeine consumers ($4.2 \pm 3.8$ servings/week).

*2.5. Testing Day Measures*

1.  State Moods: The validated POMS-SF questionnaire was utilized to determine mood states from a 5-point Likert scale as previously described [15]. The subjects in this study scored similarly with previously published literature [20].
2.  State Mental and Physical Energy and Fatigue: MPSTEF determined participants feelings. The state aspect of this scale uses a similar 12-item Visual Analog Scale (VAS). However, as performed previously [21] we utilized a modified scale scoring from 0 (No feelings) to 10 (highest imaginable feelings) to account for certain limitations in technical data collecting. Cronbach's α ranged from 0.707 to 0.874 (state physical energy (PE) = 0.785, state physical fatigue (PF) = 0.837, state mental energy (ME) = 0.707, state mental fatigue (MF) = 0.874).
3.  Serial Three and Serial Seven Subtraction Tasks: Subjects silently subtracted random numbers (ranging between 800 and 999) either by threes (SS3) or sevens (SS7) from a computer screen (Tahoma Regular font, size 20 pt). Participants were instructed to correctly answer as quickly and as accurately as possible, with allowances to maximally complete their attempts within a two minute time frame [22,23]. Only the number of attempts for the serial subtract tasks were analyzed. The average response

accuracy rate for all tests was >97.5%, suggesting that the individuals were sacrificing speed for accuracy and fatigued during this task.

4. Fine Motor Performance: The nine-hole peg test (9HPT) of finger dexterity measured fine motor performance [24]. Briefly, this test was performed with participants sequentially using their dominant hand (DH), then their non-dominant hand (NDH). Tests with each hand were alternated and performed twice. Mean test scores were averaged (measured in seconds) for both hands.

5. Physiological Measures: We measured blood pressure (BP) and heart rate (HR) as previously described [15]

### 2.6. Procedure

Familiarization Day: To minimize experimental error via learning effects, participants completed a practice mental task session.

Testing Days 1–3: Utilizing randomizer.org, the subjects were allocated randomly according to beverage order. Participants then completed a survey to determine testing day eligibility. Next, they provided a saliva sample according to a drool down method to be tested after the study to determine compliance with the caffeine pre-testing caffeine restraint. Participants who met the pre-testing criteria completed the mental task battery. After completing this battery, participants were provided the beverage they were asked to consume within 2 min and were then given a 28 min break where they were instructed not to perform any physically or cognitively challenging tasks. After the break, participants completed three more mental energy test batteries followed by 10-min breaks between each one. After the final mental testing, subjects' salivary caffeine levels were again collected as described above.

### 2.7. Data Analysis

Differences in measures across time between groups of beverage and trait were established using a linear mixed-effects model from an lme4 package in R [25]. Each model tested three null hypotheses at a significance level of 0.05. Trait moods were binned into the 50th percentile for high and low groups. There were no significant difference in the measure between beverages over time, among trait groups over time, or towards trait×beverage over time.

## 3. Results

All results are reported in Table 4.

**Table 4.** Significant Analyses of Trait Mental and Physical Status According to Test Beverages.

| Factor. | Measure | β (95% CI) | *t* Statistic | *p* Value |
|---------|---------|------------|---------------|-----------|
| TPE | Vigor | 0.087 (−1.72, 1.894) | 0.095 | 0.925 |
| Beverage | Vigor | 1.171 (−0.34, 2.683) | 1.518 | 0.130 |
| TPE × Beverage | Vigor | −1.444 (−3.94, 1.052) | −1.134 | 0.258 |
| TPF | Vigor | −0.183 (−1.942, 1.575) | −0.204 | 0.839 |
| Beverage | Vigor | 0.417 (−1.289, 2.123) | 0.479 | 0.633 |
| TPF × Beverage | Vigor | 0.45 (−1.962, 2.862) | 0.366 | 0.715 |
| TME | Vigor | −0.506 (−2.477, 1.465) | −0.503 | 0.616 |
| Beverage | Vigor | 0.75 (−0.659, 2.159) | 1.044 | 0.298 |
| TME × Beverage | Vigor | −0.406 (−3.134, 2.322) | −0.292 | 0.771 |
| TMF | Vigor | −1.147 (−3.954, 1.659) | −0.801 | 0.429 |

**Table 4.** *Cont.*

| Factor. | Measure | β (95% CI) | *t* Statistic | *p* Value |
|---|---|---|---|---|
| Beverage | Vigor | 0.5 (−0.608, 1.608) | 0.885 | 0.379 |
| TMF × Beverage | Vigor | −0.393 (−2.014, 1.228) | −0.475 | 0.636 |
| TPE | Fatigue | −0.445 (−1.974, 1.085) | −0.57 | 0.570 |
| Beverage | Fatigue | 0.342 (−0.898, 1.582) | 0.541 | 0.589 |
| TPE × Beverage | Fatigue | 0.726 (−1.321, 2.774) | 0.695 | 0.488 |
| TPF | Fatigue | 0.767 (−0.674, 2.207) | 1.043 | 0.300 |
| Beverage | Fatigue | 0.517 (−0.88, 1.914) | 0.725 | 0.469 |
| TPF × Beverage | Fatigue | 0.183 (−1.792, 2.159) | 0.182 | 0.856 |
| TME | Fatigue | −1.159 (−2.817, 0.498) | −1.371 | 0.174 |
| Beverage | Fatigue | 0.216 (−0.933, 1.365) | 0.368 | 0.713 |
| TME × Beverage | Fatigue | 1.472 (−0.753, 3.696) | 1.296 | 0.196 |
| TMF | Fatigue | 1.545 (−0.151, 3.24) | 1.786 | 0.082 |
| Beverage | Fatigue | 0.344 (−0.583, 1.271) | 0.727 | 0.469 |
| TMF × Beverage | Fatigue | 0.085 (−1.272, 1.441) | 0.122 | 0.903 |
| TPE | Tension | 0.848 (0.187, 1.509) | 2.515 | 0.014 |
| Beverage | Tension | 0.434 (−0.06, 0.928) | 1.722 | 0.086 |
| TPE × Beverage | Tension | −0.866 (−1.682, −0.05) | −2.08 | 0.039 |
| TPF | Tension | 0.117 (−0.419, 0.353) | 0.349 | 0.729 |
| Beverage | Tension | 0.267 (−0.294, 0.828) | 0.932 | 0.353 |
| TPF × Beverage | Tension | −0.3 (−1.093, 0.493) | −0.741 | 0.460 |
| TME | Tension | −0.352 (−0.803, 0.238) | −0.94 | 0.351 |
| Beverage | Tension | 0.08 (−0.384, 0.543) | 0.336 | 0.737 |
| TME × Beverage | Tension | 0.139 (−0.759, 1.037) | 0.304 | 0.762 |
| TMF | Tension | 0.192 (−0.454, 0.668) | 0.475 | 0.638 |
| Beverage | Tension | 0.063 (−0.342, 0.467) | 0.303 | 0.763 |
| TMF × Beverage | Tension | −0.17 (−0.762, 0.423) | −0.561 | 0.576 |
| TPE | PE | 0.511 (−1.65, 2.671) | 0.463 | 0.644 |
| Beverage | PE | 0.105 (−1.641, 1.851) | 0.118 | 0.906 |
| TPE × Beverage | PE | −1.492 (−4.375, 1.392) | −1.014 | 0.312 |
| TPF | PE | 1.983 (−0.027, 3.994) | 1.934 | 0.056 |
| Beverage | PE | 0.05 (−1.918, 2.018) | 0.05 | 0.960 |
| TPF × Beverage | PE | −0.983 (−3.766, 1.799) | −0.693 | 0.489 |
| TME | PE | −2.321 (−4.595, −0.048) | −2.001 | 0.048 |
| Beverage | PE | −0.784 (−2.408, 0.84) | −0.946 | 0.345 |
| TME × Beverage | PE | 1.284 (−1.861, 4.429) | 0.8 | 0.424 |
| TMF | PE | 2.509 (−0.308, 5.326) | 1.746 | 0.087 |
| Beverage | PE | 1.281 (−0.721, 3.284) | 1.254 | 0.213 |
| TMF × Beverage | PE | −2.674 (−5.606, 0.258) | −1.788 | 0.077 |
| TPE | PF | −0.976 (−3.342, 1.39) | −0.809 | 0.42 |
| Beverage | PF | 1.263 (−0.763, 3.289) | 1.222 | 0.223 |
| TPE × Beverage | PF | −1.081 (−4.427, 2.265) | −0.633 | 0.527 |
| TPF | PF | −1.3 (−3.576, 0.976) | −1.119 | 0.264 |
| Beverage | PF | −0.817 (−3.093, 1.46) | −0.703 | 0.483 |
| TPF × Beverage | PF | 3.367 (0.148, 6.586) | 2.05 | 0.041 |
| TME | PF | 0.528 (−2.066, 3.123) | 0.399 | 0.69 |
| Beverage | PF | 0.784 (−1.111, 2.679) | 0.811 | 0.418 |
| TME × Beverage | PF | 0.31 (−3.36, 3.979) | 0.165 | 0.869 |
| TMF | PF | −0.839 (−4.236, 2.558) | −0.484 | 0.63 |
| Beverage | PF | −0.719 (−3.391, 1.953) | −0.527 | 0.599 |
| TMF × Beverage | PF | 3.362 (−0.55, 7.273) | 1.684 | 0.096 |
| TPE | ME | 0.366 (−1.819, 2.551) | 0.328 | 0.743 |
| Beverage | ME | 0.329 (−1.542, 2.2) | 0.345 | 0.731 |
| TPE × Beverage | ME | −1.42 (−4.51, 1.67) | −0.901 | 0.369 |
| TPF | ME | 1.167 (−0.938, 3.271) | 1.087 | 0.278 |
| Beverage | ME | 0.55 (−1.554, 2.654) | 0.512 | 0.609 |
| TPF × Beverage | ME | −1.483 (−4.459, 1.493) | −0.977 | 0.33 |
| TME | ME | −1.849 (−4.221, 0.523) | −1.528 | 0.128 |
| Beverage | ME | −0.477 (−2.21, 1.255) | −0.54 | 0.59 |

**Table 4.** *Cont.*

| Factor. | Measure | β (95% CI) | *t* Statistic | *p* Value |
|---|---|---|---|---|
| TME × Beverage | ME | 1.071 (−2.284, 4.426) | 0.626 | 0.532 |
| TMF | ME | 2.103 (−1.109, 5.314) | 1.283 | 0.205 |
| Beverage | ME | 1.812 (−0.562, 4.187) | 1.496 | 0.138 |
| TMF × Beverage | ME | −3.777 (−7.253, −0.3) | −2.129 | 0.036 |
| TPE | MF | −2.068 (−4.39, 0.253) | −1.746 | 0.084 |
| Beverage | MF | 0.75 (−1.153, 2.653) | 0.772 | 0.441 |
| TPE × Beverage | MF | −0.341 (−3.484, 2.803) | −0.213 | 0.832 |
| TPF | MF | −0.383 (−2.712, 1.945) | −0.323 | 0.748 |
| Beverage | MF | −0.9 (−3.022, 1.222) | −0.831 | 0.407 |
| TPF × Beverage | MF | 3.05 (0.048, 6.052) | 1.992 | 0.048 |
| TME | MF | −2.119 (−4.769, 0.53) | −1.568 | 0.121 |
| Beverage | MF | 0.102 (−1.662, 1.866) | 0.114 | 0.91 |
| TME × Beverage | MF | 1.96 (−1.455, 5.376) | 1.125 | 0.262 |
| TMF | MF | 0.094 (−3.142, 3.33) | 0.057 | 0.955 |
| Beverage | MF | 0.344 (−2.105, 2.793) | 0.275 | 0.784 |
| TMF × Beverage | MF | 1.871 (−1.714, 5.456) | 1.023 | 0.309 |
| TPE | SS3 | 1.813 (−5.279, 8.906) | 0.501 | 0.62 |
| Beverage | SS3 | −1.342 (−2.799, 0.114) | −1.806 | 0.072 |
| TPE × Beverage | SS3 | 2.706 (0.301, 5.111) | 2.205 | 0.029 |
| TPF | SS3 | 3.467 (−3.302, 10.235) | 1.004 | 0.324 |
| Beverage | SS3 | −0.9 (−2.555, 0.755) | −1.066 | 0.288 |
| TPF × Beverage | SS3 | 1.1 (−1.24, 3.44) | 0.921 | 0.358 |
| TME | SS3 | 1.744 (−6.044, 9.533) | 0.439 | 0.664 |
| Beverage | SS3 | −0.659 (−2.026, 0.708) | −0.945 | 0.346 |
| TME × Beverage | SS3 | 1.159 (−1.488, 3.806) | 0.858 | 0.392 |
| TMF | SS3 | 4.156 (−2.557, 10.87) | 1.213 | 0.235 |
| Beverage | SS3 | −0.953 (−2.554, 0.648) | −1.167 | 0.245 |
| TMF × Beverage | SS3 | 1.292 (−1.051, 3.636) | 1.081 | 0.281 |
| TPE | SS7 | −0.914 (−6.885, 5.058) | −0.3 | 0.766 |
| Beverage | SS7 | −0.303 (−1.6, 0.994) | −0.457 | 0.648 |
| TPE × Beverage | SS7 | 1.166 (−0.976, 3.308) | 1.067 | 0.287 |
| TPF | SS7 | 1.633 (−4.072, 7.339) | 0.561 | 0.579 |
| Beverage | SS7 | −0.317 (−1.778, 1.145) | −0.425 | 0.671 |
| TPF × Beverage | SS7 | 0.883 (−1.183, 2.95) | 0.838 | 0.403 |
| TME | SS7 | 2.256 (−4.168, 8.679) | 0.688 | 0.497 |
| Beverage | SS7 | −0.182 (−1.388, 1.024) | −0.296 | 0.768 |
| TME × Beverage | SS7 | 1.151 (−1.185, 3.486) | 0.966 | 0.335 |
| TMF | SS7 | 2.75 (−2.899, 8.399) | 0.954 | 0.348 |
| Beverage | SS7 | −0.266 (−1.681, 1.15) | −0.368 | 0.713 |
| TMF × Beverage | SS7 | 0.837 (−1.234, 2.909) | 0.792 | 0.429 |
| TPE | NDH | 7.194 (−4.787, 19.175) | 1.177 | 0.249 |
| Beverage | NDH | 1.934 (−0.594, 4.462) | 1.5 | 0.135 |
| × Beverage | NDH | −7.298 (−11.472, −3.123) | −3.426 | 0.001 |
| TPF | NDH | 0.733 (−10.831, 12.298) | 0.124 | 0.902 |
| Beverage | NDH | 2.917 (0.078, 5.755) | 2.014 | 0.045 |
| TPF × Beverage | NDH | −7.317 (−11.331, −3.303) | −3.573 | 0 |
| TME | NDH | 5.713 (−7.261, 18.687) | 0.863 | 0.395 |
| Beverage | NDH | −0.739 (−3.153, 1.676) | −0.6 | 0.549 |
| TME × Beverage | NDH | −0.011 (−4.687, 4.664) | −0.005 | 0.996 |
| TMF | NDH | −4.324 (−15.625, 6.978) | −0.75 | 0.459 |
| Beverage | NDH | 2.281 (−0.485, 5.048) | 1.616 | 0.108 |
| TMF × Beverage | NDH | −6.478 (−10.528,) | −3.135 | 0.002 |
| TPE | DH | 3.539 (−8.145, 15.224) | 0.594 | 0.557 |
| Beverage | DH | 0.921 (−1.645, 3.487) | 0.704 | 0.482 |
| TPE × Beverage | DH | −0.33 (−4.567, 3.907) | −0.153 | 0.879 |
| TPF | DH | 1.85 (−9.468, 13.168) | 0.32 | 0.751 |
| Beverage | DH | 3.3 (0.452, 6.148) | 2.271 | 0.024 |

**Table 4.** *Cont.*

| Factor. | Measure | β (95% CI) | *t* Statistic | *p* Value |
|---|---|---|---|---|
| TPF × Beverage | DH | −5 (−9.027, −0.973) | −2.434 | 0.016 |
| TME | DH | 6.125 (−6.456, 18.706) | 0.954 | 0.348 |
| Beverage | DH | 0.625 (−1.759, 3.009) | 0.514 | 0.608 |
| TME × Beverage | DH | 0.656 (−3.961, 5.273) | 0.279 | 0.781 |
| TMF | DH | −1.924 (−13.167, 9.318) | −0.335 | 0.74 |
| Beverage | DH | 2.875 (0.109, 5.641) | 2.038 | 0.043 |
| TMF × Beverage | DH | −4.446 (−8.495, −0.398) | −2.153 | 0.032 |
| TPE | SBP | −1.604 (−8.781, 5.573) | −0.438 | 0.664 |
| Beverage | SBP | 0.25 (−1.323, 1.823) | 0.312 | 0.756 |
| TPE × Beverage | SBP | −2.659 (−5.256, −0.062) | −2.007 | 0.046 |
| TPF | SBP | −0.817 (−7.813, 6.18) | −0.229 | 0.821 |
| Beverage | SBP | −1.717 (−3.493, 0.06) | −1.894 | 0.06 |
| TPF × Beverage | SBP | 1.983 (−0.529, 4.496) | 1.547 | 0.123 |
| TME | SBP | 0.784 (−7.127, 8.695) | 0.194 | 0.847 |
| Beverage | SBP | −0.17 (−1.639, 1.298) | −0.228 | 0.82 |
| TME × Beverage | SBP | −2.08 (−4.923, 0.764) | −1.434 | 0.153 |
| TMF | SBP | 0.308 (−6.691, 7.307) | 0.086 | 0.932 |
| Beverage | SBP | −1.562 (−3.285, 0.16) | −1.778 | 0.077 |
| TMF × Beverage | SBP | 1.795 (−0.726, 4.316) | 1.395 | 0.164 |
| TPE | DBP | −5.343 (−9.867, −0.82) | −2.315 | 0.027 |
| Beverage | DBP | 0.408 (−0.866, 1.682) | 0.627 | 0.531 |
| TPE × Beverage | DBP | 0.547 (−1.557, 2.651) | 0.509 | 0.611 |
| TPF | DBP | 1.15 (−3.568, 5.868) | 0.478 | 0.636 |
| Beverage | DBP | 1.6 (0.178, 3.022) | 2.205 | 0.029 |
| TPF × Beverage | DBP | −1.983 (−3.994, 0.028) | −1.933 | 0.055 |
| TME | DBP | −7.551 (−12.128, −2.974) | −3.234 | 0.003 |
| Beverage | DBP | 0.534 (−0.65, 1.719) | 0.884 | 0.378 |
| TME × Beverage | DBP | 0.278 (−2.016, 2.572) | 0.238 | 0.812 |
| TMF | DBP | 1.185 (−3.542, 5.912) | 0.491 | 0.627 |
| Beverage | DBP | 1.312 (−0.07, 2.695) | 1.861 | 0.064 |
| TMF × Beverage | DBP | −1.509 (−3.532, 0.514) | −1.462 | 0.145 |
| TPE | HR | −1.758 (−7.207, 3.69) | −0.632 | 0.531 |
| Beverage | HR | 0.118 (−1.749, 1.986) | 0.124 | 0.901 |
| TPE × Beverage | HR | 0.45 (−2.634, 3.533) | 0.286 | 0.775 |
| TPF | HR | −0.3 (−5.548, 4.948) | −0.112 | 0.911 |
| Beverage | HR | −1.433 (−3.509, 0.642) | −1.353 | 0.177 |
| TPF × Beverage | HR | 3.433 (0.498, 6.369) | 2.292 | 0.023 |
| TME | HR | −4.006 (−9.854, 1.843) | −1.342 | 0.189 |
| Beverage | HR | −0.136 (−1.868, 1.596) | −0.154 | 0.878 |
| TME × Beverage | HR | 1.574 (−1.78, 4.928) | 0.92 | 0.359 |
| TMF | HR | −1.272 (−6.561, 4.017) | −0.472 | 0.64 |
| Beverage | HR | −0.813 (−2.836, 1.211) | −0.787 | 0.432 |
| TMF × Beverage | HR | 2.348 (−0.614, 5.31) | 1.554 | 0.122 |

PE = State physical energy, PF = State physical fatigue, ME = State mental energy, MF = State mental fatigue, SS3 = serial 3 subtraction, SS7 = serial 7 subtraction, NDH = Non-dominant hand average time, DH = Dominant hand average time, SBP = systolic blood pressure, DBP = diastolic blood pressure, HR = heart rate.

### 3.1. Moods and Cognitive Measures

The results of the linear mixed-effects models are shown in Table 3. There was no significant difference in tension between beverages; however, those with initially higher TPE reported significantly increased tension (β = 0.848, *p* = 0.014) over time than those with lower TPE. Moreover, those with lower TPE saw lower tension with the e+shot (β = −0.866, *p* = 0.039) than the caffeinated placebo, while those with higher TPE saw higher tension with the caffeinated placebo when compared to the e+shot. PE was significantly lower (β = −2.321, *p* = 0.048) in the high TME category when controlling for beverage type. Neither beverage type nor TPF category independently impacted levels of PF; however, their interaction effect was statistically significant (β = 3.367, *p* = 0.041) (Figure 1A). While

low TPF individuals saw little to no change in PF between categories of beverage, those with high TPF experienced much higher PF with the caffeinated placebo than the e+shot. This trend was similar to TPF and MF, with no significant differences across time except for the interaction effect ($\beta$ = 3.05, *p* = 0.048), with low TPF individuals seeing little change in MF with the caffeinated placebo to the e+shot (Figure 1B). High TPF individuals experienced considerably more MF from the caffeinated placebo than from the e+shot. SS3 saw a significant crossover effect between low and high TPE and beverage type. While no significant differences were present between beverage types or between low and high TPE, e+shot appeared to slightly increase performance for the SS3 task when compared to the caffeinated placebo in low-TPE individuals. In contrast, high-TPE individuals appeared to do better over time after consuming the caffeinated placebo than e+shot (Table 4).

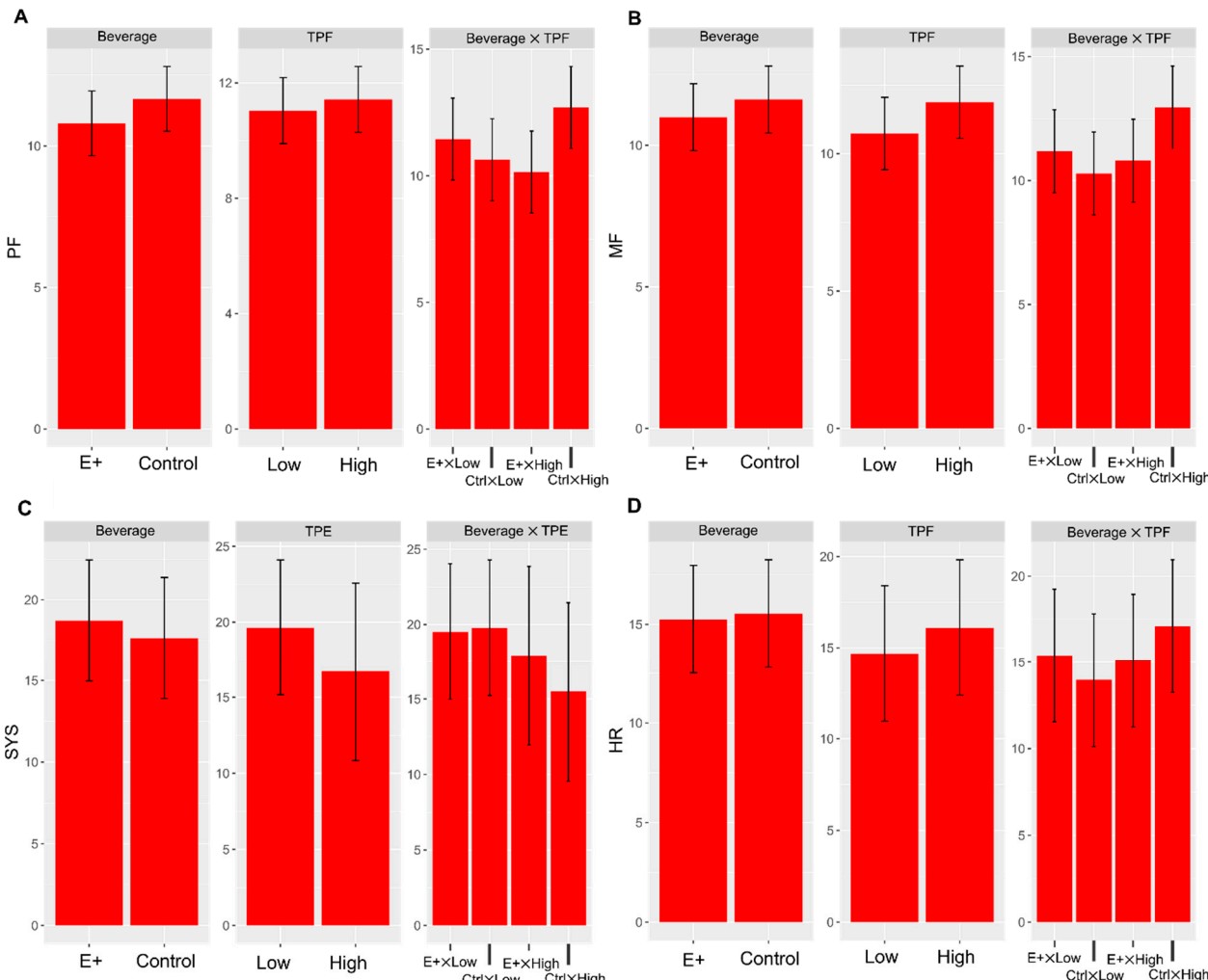

**Figure 1.** Marginal means over time and standard errors for each group for 4 different models. Both PF (**A**) and MF (**B**) are examples of higher-order interaction effects between beverage type and TPF. Similar interaction effects are shown for systolic blood pressure (**C**) and heart rate (**D**), related to TPE and TPF, respectively.

### 3.2. Fine Motor Performance and Physiologic Measures

The 9HPT saw a significant crossover effect ($\beta$ = 2.706, *p* = 0.029) for the DH between beverage type and TPF ($\beta$ = −5.000, *p* = 0.016) and TMF ($\beta$ = −4.446, *p* = 0.032). For TMF and TPF, the low fatigue groups saw quicker completion times with the e+shot than the caffeinated placebo, while there was a negligible difference in completion time between beverages for the high fatigue groups. There were also significant crossover effects for the 9HPT for the NDH between beverage type and TPE ($\beta$ = −7.298, *p* < 0.001), with the beverage

type making no difference for the low TPE group while the e+shot significantly outperformed the caffeinated placebo for the high TPE group. Additionally, TPF (β = −7.317, *p* < 0.001) and TMF (β = −6.478, *p* = 0.002) scores indicated that low-fatigue individuals consuming e+shot outperformed low-fatigue individuals consuming the caffeinated placebo; however, the opposite results were observed for the high fatigue groups.

High TPE exhibited a statistically significant crossover effect (β = −2.659, *p* = 0.046) with systolic BP (SBP) and beverage type, with the low TPE group exhibiting no difference between beverages (Figure 1C). Still, the high TPE group exhibited a greater reduction in SBP from the caffeinated placebo than e+shot. Both high TPE (β = −5.343, *p* = 0.027) and high TME (β = −7.551, *p* = 0.003) were significantly associated with decreased diastolic BP (DBP) over time. Heart rate measures revealed a significant crossover effect between TPF and beverage type (β = 3.430, *p* = 0.023), with the low TPF group exhibiting no significant change in HR while the high TPF group significantly increased HR in response to the caffeinated placebo compared with no increase for the e+shot (Figure 1D).

A summary of all results can be found in Supplemental Figures S1–S4.

## 4. Discussion

The purpose of this post-hoc analysis was to re-examine previously published data [15] to determine whether an individual's trait mood (long-standing predisposition to a mood) status contributed to inter-individual differences observed between acute consumption of a caffeinated beverage or a beverage containing similar caffeine content and adaptogenic herbs (e+shot) on neurocognitive outcomes. Our analysis suggests that TME, TPE, TMF, and TPF were uniquely modified by e+shot when compared to caffeine, with e+shot selectively benefiting individuals with low TPE or high TPF. However, e+shot, similar to the caffeinated placebo, only benefitted individuals characterized by high energy/low fatigue on fine motor task performance. Interestingly, subjects characterized by high TPE revealed a more significant decline in SBP after consuming the caffeinated placebo than after consuming the e+shot. Additionally, low TPF individuals had a higher HR after consuming the e+shot than after consuming the caffeinated placebo, while high TPF individuals responded with a lower HR after consuming the e+shot and a higher HR after consuming the caffeinated placebo. Overall, these results suggest that individuals experiencing typically low energy/high fatigue most benefitted from consuming both caffeine and adaptogenic herbs.

The findings of this analysis are unique and leave very little comparable literature allowing us to examine previous findings more critically. For example, our initial publication [15] suggested no significant difference in anxiety among beverages. However, upon stratifying trait status, we now observed that caffeinated placebo increased anxiety in all participants. In contrast, those with low TPE reported significantly less increases in anxiety with the consumption of e+shot, whereas those with high TPE reported considerably more anxiety when consuming e+shot than caffeinated placebo. These results suggest that some individuals were resistant to the anxiety-provoking effects of caffeine when consuming e+shot. Indeed, anti-anxiety, anti-stress, anti-mental fatigue properties have long been ascribed to adaptogenic herbs [25]. In Copley et al. [26], supplementation with rhodiola for two weeks reduced anxiety in students characterized as mildly anxious (low trait anxiety) according to the Spielberger-State-Trait-Anxiety-Inventory.

Our findings also suggest that TME, TPE, and TPF uniquely influence objective and subjective indices of mental energy and fatigue. For example, we report that individuals who are high TPF report the greatest benefits of e+shot on state mental fatigue, while individuals who are high TPE report the greatest benefit of consuming caffeine on the serial subtraction three task. Our findings may explain the inconsistent findings of studies assessing the effects of adaptogenic herbs on neurocognitive performance. For example, several adaptogenic herbs in e+shot, such as *Eleutherococcus senticosus*, *Rhodiola rosea*, and Hawthorn (*Crataegus oxyacantha*), have demonstrated clinical benefits on neurocognitive performance in many studies [26–28], whereas others have found no effects [29]. Mechanistically, one

potential rationale for adaptogens' influence on psychosocial stress and neurocognitive measures could be via down-regulation of G-protein-coupled-receptor-signaling pathways, such as those modulating neurotransmitter activation and the HPA axis [30]. The potential benefits of adaptogenic herbs on many of these outcomes have recently been compiled by Gerontakos et al. [31], who also noted a potential rationale for the heterogeneity reported in the scientific literature.

Another interesting finding is that we reported no significant differences between beverages on feelings of energy and fatigue [15]. However, our post-hoc analysis reports that e+shot effectively reduced feelings of mental and physical fatigue in high-TPF individuals compared to the caffeinated placebo. Additionally, individuals with low TME also reported increased feelings of physical energy when consuming e+shot, rather than the caffeinated placebo. Again, these findings suggest that e+shot benefited those who typically were low energy/high fatigue, more so than the caffeinated placebo.

While e+shot provided ME and MF benefits to individuals who usually feel low energy/high fatigue, its impact on fine motor task performance was the opposite. Our findings suggest that for the NDH, e+shot significantly improved performance for individuals who reported being low TMF and TPF, while e+shot reduced performance for those who were high TPE. Interestingly, the caffeinated placebo reduced performance for those who were low TPF and low TMF.

Although within normal ranges, a statistically significant increase in BP was reported [15] with both beverages, with e+shot reporting a lesser increase than caffeine alone. After reevaluating the data, we observed no significant SBP differences among beverages for the low TPE subjects. However, high TPE was associated with significantly attenuated increases in SBP when consuming the caffeinated placebo when compared to e+shot. Additionally, for those exhibiting low TPE and TME, both beverages significantly increased DBP when compared to high TPE or TME individuals. Although none of our participants reported being hypertensive after consuming either beverage, our findings suggest that those who typically report low TME or TPE should monitor potential caffeine-induced elevated blood pressure.

This post-hoc analysis also reveals that the caffeinated beverage increased HR for those who typically feel physically fatigued (high TPF) more than those who do not. However, there were no significant differences between groups for e+shot. These findings suggest that caffeine by itself may have an increased physiological effect on non-habitual drinkers of caffeine who typically feel physically fatigued.

With an increased focus on personalized nutrition [32,33], this study adds to the literature by providing a potential screening tool for identifying individual neurocognitive or physiological responses to caffeine [14,34–37]. Additionally, this may raise the possibility that various forms and delivery of caffeine may be relevant when investigating inter-individual responses. Moreover, researchers interested in understanding these differences may also consider how trait level mental and physical energy and fatigue status are modified under specific conditions (i.e., sports performance) when identifying hyper-, hypo-, and non-responders to caffeinated beverages. Finally, we recommend that researchers interested in understanding interindividual differences in neurocognitive outcomes with various nutritional interventions should also consider examining trait level energy and fatigue to explain inter-individual differences.

## 5. Conclusions

The objective of this post-hoc analysis was to determine whether a subject's longstanding pre-disposition to feelings of energy and fatigue modified the effects of caffeine alone or the combination of caffeine plus adaptogenic herbs on indices of mental energy and mental fatigue, moods, and physiological responses. Our findings suggest that the adaptogenic rich beverage was most beneficial for individuals who typically reported feeling low mental energy and high mental fatigue and was also beneficial in attenuating caffeine's anxiety-provoking effects in individuals who report low physical energy. Future researchers

should account for trait energy and fatigue as they may modify the effect of their nutritional intervention on acute changes in indices of mental energy and mental fatigue.

**Supplementary Materials:** The following supporting information can be downloaded at: https://www.mdpi.com/article/10.3390/app12094466/s1, Figure S1: Trait Mental Energy; Figure S2: Trait Mental Fatigue; Figure S3: Trait Physical Energy; Figure S4 Trait Physical Fatigue.

**Author Contributions:** A.B. and D.F. were responsible for the study design, data interpretation, writing of the first and final draft. D.F. and S.M. were responsible for performing all analyses. A.B. and D.F. were responsible for interpreting all statistical analyses. A.B. was responsible for data collection. S.M. and E.G. were responsible for the writing of the first and final draft. All authors have read and agreed to the published version of the manuscript.

**Funding:** The original study was funded by a grant from Isagenix International, LLC. The funders had no role in study design, data collection, the re-analysis of the data, or the decision to publish the manuscript.

**Institutional Review Board Statement:** Approval for this study was granted by the Clarkson University Institutional Review Board (approval# 16-34.1).

**Informed Consent Statement:** All participants read and signed an informed consent form.

**Data Availability Statement:** The dataset supporting the conclusions of this article is available in the Mendeley respository. The DOI number for this dataset is 10.17632/3s8sr9zth9.1.

**Acknowledgments:** The authors would like to acknowledge undergraduate researchers Stephanie Grobe and Joanne tiRiele for their help in data collection.

**Conflicts of Interest:** E.G. is currently Director of Research and Science at Isagenix International, LLC, the sponsor of the original study. Author A.B. received funding to conduct the original study. The remaining authors declare no conflict of interest.

**Clinical Registration:** This trial was registered at clinicaltrials.gov (NCT03850275)—retrospectively registered 20 February 2019. https://clinicaltrials.gov/ct2/show/NCT03850275?term=The+Effects+of+e%2BShots+Energy+Beverage+on+Mental+Energy&draw=2&rank=1 (accessed on 25 April 2022).

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
