# Peer review of "Trait Energy and Fatigue Modify Acute Ingestion of an Adaptogenic-Rich Beverage on Neurocognitive Performance"

_applsci, doi:10.3390/app12094466_

Round 1
Reviewer 1 Report
Continuing the previous work where the acute effects a naturally-sourced caffeine and adaptogenic-rich energy shot were compared on a variety of mental performance and cognitive parameters in young adults, in this paper the authors examine whether a subject’s long-standing pre-disposition to feelings of energy and fatigue can modify the effects of caffeine alone or the combination of caffeine plus adaptogenic herbs. The experiments were set properly, the results are clearly presented and the conclusions are supported by the results.
Author Response
Dear Reviewer,
We would like to thank the reviewer for taking the time to review our manuscript and for providing us feedback. Below are your comments. Our responses can be found in red.
Reviewer 1
Continuing the previous work where the acute effects a naturally-sourced caffeine and adaptogenic-rich energy shot were compared on a variety of mental performance and cognitive parameters in young adults, in this paper the authors examine whether a subject’s long-standing pre-disposition to feelings of energy and fatigue can modify the effects of caffeine alone or the combination of caffeine plus adaptogenic herbs. The experiments were set properly, the results are clearly presented and the conclusions are supported by the results.
We would like to thank the reviewer for their affirming words.

Reviewer 2 Report
The manuscript entitled "Trait Energy and Fatigue Modify Acute Ingestion of an Adaptogenic-rich Beverage on Neurocognitive Performance" is well written and described. This study showed that the adaptogenic rich beverages help to restore mood, mental fatigue, and other physiological responses, and the presented data support this hypothesis.
Author Response
Dear Reviewer,
We would like to thank the reviewer for taking the time to review our manuscript and for providing us feedback. Below are your comments. Our responses can be found in red.
Reviewer 2
The manuscript entitled "Trait Energy and Fatigue Modify Acute Ingestion of an Adaptogenic-rich Beverage on Neurocognitive Performance" is well written and described. This study showed that the adaptogenic rich beverages help to restore mood, mental fatigue, and other physiological responses, and the presented data support this hypothesis.
We would like to thank the reviewer for their affirming words

Reviewer 3 Report
In this submission, Ali Boolani and colleagues performed a post-hoc re-analysis of a previous study of theirs on the effects of an adaptogen-rich, caffeine-containing product on mood and cognitive tests as compared to a caffeine-matched placebo. Such re-analysis stratified the participants by initial trait mental and physical energy and trait mental and physical fatigue status. The Authors found that the consumption of such product preferentially benefited individuals with initial low trait mental and physical energy and high trait mental fatigue compared to caffeine alone, also attenuating the anxiety-provoking effects of caffeine in individuals who report low physical energy. This would suggest that adaptogens may promote mental and physical performance benefits while modulating potentially negatively associated responses to caffeine.
I have no major concerns regarding this piece of research, but I have some suggestions:
- The Abstract takes it too much for granted that the reader is aware of the previous work to which the Authors refer. I would suggest a revision to make it easier for the naïve reader to understand the paper.
- To facilitate reading for those who are not familiar with adaptogens and the e+Energy Shot, I suggest that the Authors provide more background in this regard.
- The total number of participants (N=30) appears only in the Abstract but not in the main text. Please fix this issue.
- Do the Authors think that their study could have – in the more or less distant future – any implication in terms of treatment/support of mental disorders? Please discuss this topic in the context of existing research on nutrient supplements and repurposed drugs. I strongly suggest that the Authors refer to these two recent relevant papers: Firth et al., 2019 (https://doi.org/10.1002/wps.20672); Bartoli et al., 2021 (https://doi.org/10.1016/j.jpsychires.2021.09.018).
Author Response
Dear Reviewer,
We would like to thank the reviewer for taking the time to review our manuscript and for providing us feedback that we feel will make our manuscript significantly stronger. Below are your comments. Our responses can be found in red.
Reviewer 3
In this submission, Ali Boolani and colleagues performed a post-hoc re-analysis of a previous study of theirs on the effects of an adaptogen-rich, caffeine-containing product on mood and cognitive tests as compared to a caffeine-matched placebo. Such re-analysis stratified the participants by initial trait mental and physical energy and trait mental and physical fatigue status. The Authors found that the consumption of such product preferentially benefited individuals with initial low trait mental and physical energy and high trait mental fatigue compared to caffeine alone, also attenuating the anxiety-provoking effects of caffeine in individuals who report low physical energy. This would suggest that adaptogens may promote mental and physical performance benefits while modulating potentially negatively associated responses to caffeine.
I have no major concerns regarding this piece of research, but I have some suggestions:
- The Abstract takes it too much for granted that the reader is aware of the previous work to which the Authors refer. I would suggest a revision to make it easier for the naïve reader to understand the paper.
We have made a few modifications to the Abstract to simplify its understanding and comprehension. If the reviewer would like us to make additional changes, we would be more than happy to make them.
- To facilitate reading for those who are not familiar with adaptogens and the e+Energy Shot, I suggest that the Authors provide more background in this regard.
Thank you for this opportunity. We have provided some additional information on adaptogens in the Introduction. The adaptogenic blend formulated for e+shot closely resembles blends utilized by Dr. Israel Brekhman, considered the “Father of adaptogens” (personal communications from co-author EG). Dr. Brekhman was a Russian scientist who first systematically studied adaptogens. We have included Brekhman’s first English-language review in the published literature among the additional information provided in the Introduction.
- The total number of participants (N=30) appears only in the Abstract but not in the main text. Please fix this issue.
We’d like to thank the reviewer for pointing this out. We have addressed this in the screening and participants section. The section states
“Characteristics of the thirty (women = 17, men = 13) participants, including all the post-hoc analyses are reported in Table 3”
- Do the Authors think that their study could have – in the more or less distant future – any implication in terms of treatment/support of mental disorders? Please discuss this topic in the context of existing research on nutrient supplements and repurposed drugs. I strongly suggest that the Authors refer to these two recent relevant papers: Firth et al., 2019 (https://doi.org/10.1002/wps.20672); Bartoli et al., 2021 (https://doi.org/10.1016/j.jpsychires.2021.09.018).
We want to thank the reviewer for pointing us to these papers. After reading these studies, because we did not test a clinical population, we cannot make any inferences about how our study, or our interventions, might impact such a population. Instead, we feel that this study identifies a low-cost measurement tool that may help us identify inter-individual differences in responses to two different caffeinated beverages. While significant evidence exists on how various epigenomic markers influence the effects of caffeine on neurocognitive parameters (Loy, et al, 2015, Childs, et al 2008, Grgic, et al, 2020, Fulton, et al, 2018, Bodeman, et al, 2012), it is difficult to perform epigenetic analyses on all individuals to determine their response to caffeine. However, our study adds two very important elements to the literature regarding future implications. We provide a low-cost/no-cost method of evaluating how someone might respond to caffeine and, secondly, we provide evidence that the inter-individual responses to caffeine also vary between the types of caffeine consumed, suggesting that some people may benefit more from an adaptogen infused caffeinated beverage. In contrast, others might benefit more from synthetically sourced caffeinated beverages. As we move towards a world of personalized nutrition (Guest, et al, 2019, Pickering and Kiely, 2018), our study provides support for individuals seeking to find more cost-effective ways of identifying hyper-, hypo-, and non-responders to caffeine among young, healthy individuals without any mental health conditions. We have now added a paragraph in the discussion section about the implications of the study and future direction.
Loy, B. D., O'Connor, P. J., Lindheimer, J. B., & Covert, S. F. (2015). Caffeine is ergogenic for adenosine A2A receptor gene (ADORA2A) T allele homozygotes: a pilot study. Journal of Caffeine Research, 5(2), 73-81.
Grgic, J., Pickering, C., Bishop, D. J., Del Coso, J., Schoenfeld, B. J., Tinsley, G. M., & Pedisic, Z. (2020). ADORA2A C allele carriers exhibit ergogenic responses to caffeine supplementation. Nutrients, 12(3), 741.
Fulton, J. L., Dinas, P. C., Carrillo, A. E., Edsall, J. R., Ryan, E. J., & Ryan, E. J. (2018). Impact of genetic variability on physiological responses to caffeine in humans: A systematic review. Nutrients, 10(10), 1373.
Childs E, Hohoff C, Deckert J, Xu K, Badner J, De Wit H. Association between ADORA2A and DRD2 polymorphisms and caffeine-induced anxiety. Neuropsychopharmacology. 2008;33(12):2791–800.
Bodenmann S, Hohoff C, Freitag C, Deckert J, Rétey JV, Bachmann V, et al. Polymorphisms of ADORA2A modulate psychomotor vigilance and the effects of caffeine on neurobehavioural performance and sleep EEG after sleep deprivation. British journal of pharmacology. 2012;165(6):1904–13
Guest, N. S., Horne, J., Vanderhout, S. M., & El-Sohemy, A. (2019). Sport nutrigenomics: personalized nutrition for athletic performance. Frontiers in nutrition, 6, 8.
Pickering, C., & Kiely, J. (2018). Are the current guidelines on caffeine use in sport optimal for everyone? Inter-individual variation in caffeine ergogenicity, and a move towards personalised sports nutrition. Sports Medicine, 48(1), 7-16.
